# How Is Peripheral Injury Signaled to Satellite Glial Cells in Sensory Ganglia?

**DOI:** 10.3390/cells11030512

**Published:** 2022-02-01

**Authors:** Menachem Hanani

**Affiliations:** Laboratory of Experimental Surgery, Faculty of Medicine, Hadassah-Hebrew University Medical Center, Mount Scopus, Jerusalem 91240, Israel; hananim@cc.huji.ac.il

**Keywords:** sensory ganglia, neurons, satellite glial cells, gliosis, axonal transport, pain, nitric oxide

## Abstract

Injury or inflammation in the peripheral branches of neurons of sensory ganglia causes changes in neuronal properties, including excessive firing, which may underlie chronic pain. The main types of glial cell in these ganglia are satellite glial cells (SGCs), which completely surround neuronal somata. SGCs undergo activation following peripheral lesions, which can enhance neuronal firing. How neuronal injury induces SGC activation has been an open question. Moreover, the mechanisms by which the injury is signaled from the periphery to the ganglia are obscure and may include electrical conduction, axonal and humoral transport, and transmission at the spinal level. We found that peripheral inflammation induced SGC activation and that the messenger between injured neurons and SGCs was nitric oxide (NO), acting by elevating cyclic guanosine monophosphate (cGMP) in SGCs. These results, together with work from other laboratories, indicate that a plausible (but not exclusive) mechanism for neuron-SGCs interactions can be formulated as follows: Firing due to peripheral injury induces NO formation in neuronal somata, which diffuses to SGCs. This stimulates cGMP synthesis in SGCs, leading to their activation and to other changes, which contribute to neuronal hyperexcitability and pain. Other mediators such as proinflammatory cytokines probably also contribute to neuron-SGC communications.

## 1. Introduction

The effects of injury to peripheral nerves on neuronal cell bodies in the peripheral and central nervous systems has been a major research topic because it is related to both important clinical and basic biological questions. For example, Wallerian degeneration has been thoroughly investigated for many years, which yielded important insights into nerve regeneration and degeneration mechanisms [1]. A common approach to studying this topic is to injure the axons of spinal motor neurons or of sensory neurons in the dorsal root ganglia (DRG) or trigeminal ganglia (TG), and to follow changes in neurons and other cells [2,3]. Changes in the somata of spinal cord neurons include synaptic displacement, early microglia activation and a later astrocyte activation. Peripheral injury also induces numerous alterations in sensory ganglia, in particular in the sensory neurons and the satellite glial cells (SGCs) that surround them [4,5]. These changes include augmented neuronal firing and activation (gliosis) of SGCs. The questions that will be addressed in this article are: what are the possible pathways between the periphery and the ganglionic neurons, and what is the signaling mechanism between the neurons and the SGCs? The discussion below is an attempt to combine the available information with some novel ideas.

## 2. Assays for Changes in Neurons and SGCs

The most commonly used assay for glial activation is the upregulation of glial fibrillary acidic protein (GFAP), which is a component of intermediate filaments and which has structural and possibly other roles [6]. The timing of the GFAP upregulation from the moment of injury is an important parameter because it can provide an idea about the mechanisms that connect nerve injury to glial changes. Unlike in astrocytes, in SGCs of sensory ganglia the resting level of GFAP in SGCs is very low, making this method highly convenient. In most studies, GFAP upregulation in SGCs occurs within 2–48 h after the injury [7,8]. The timing of changes in sensory ganglia following peripheral injury can be measured most accurately in the context of post-surgical pain, because the time and location of the injury are well defined. In a study on plantar incision in rats it was found that GFAP was upregulated in SGCs of mouse DRG as early as 30 min after the incision, and declined to baseline at days 7–14 [9]. Clearly, there must be a delay between the arrival of the injury signals to the neuronal somata and the time when the level of GFAP in SGCs is sufficiently high to be measured. Therefore, it can be safely assumed that the injury signals reach the neurons within a few minutes or even seconds.

Another relevant molecule for assessing the rate of signal transmission between the periphery and sensory ganglia is the enzyme extracellular signal-regulated kinase (ERK). Zhuang et al. [10] detected phosphorylated ERK (pERK) upregulation 10 min after spinal nerve ligation (SNL) of the sciatic nerve in both the DRG and dorsal horn of the spinal cord. pERK was present in SGCs in control mice and was elevated (moderately) in these cells only from day 10. Dai et al. [11] stimulated the peripheral end of a rat sciatic nerve electrically, chemically and physiologically, and detected an increased level of pERK in DRG neurons within 2 min (which declined to baseline after 20 min). Measuring such a brief delay was achieved by vascular perfusion with a fixative immediately after making the incision. Such a short timescale puts clear constraints on the possible mechanism by which signals from a periphery are transmitted to the soma. In a more recent study, a pERK increase was observed in both neurons and SGCs within 2 min after a plantar incision in the hind paw of rats [12]. The most likely mechanism for this fast signaling is that nerve discharge caused by axonal injury reaches the neuronal somata within seconds and induces biochemical changes in them within a few minutes.

It should be kept in mind that peripheral injury may induce multiple types of changes in the axons and that each may reach the ganglia by a different mechanism and pathway and may be mediated by different signals. The possible mechanisms are described below.

## 3. Pathways from the Periphery to Sensory Ganglia

### 3.1. Electrical Conduction

Axonal conductance of action potentials is the fastest known means for signal propagation in the body. For axons of neurons in sensory ganglia, the conduction velocity of the fastest (Aα) fibers is 33–55 m/s, and for the slowest fibers (C) it is less than 1.4 m/s [13]. Thus, if we use a representative value of 0.5 m/s for slow fibers and a distance of 10 cm from the periphery to the DRG, electrical signals will arrive from the periphery to the ganglia in less than 1 s. It was shown that injury induced a barrage of electrical activity in the axons, known as “injury potential” [14,15]. Therefore, somata of sensory neurons are expected to fire excessively within seconds after the injury. One could propose that this augmented firing could lead to the activation of SGCs that surround these neurons and possibly those that surround non-injured neurons. Xie et al. [16] found that the early blockade of nerve activity reduced the subsequent pain behavior following two types of nerve injury in rats, SNL and spared nerve injury (SNI). In a later study, Xie et al. [8] tested the idea that SGC activation following injury depended on the neuronal electrical activity. They used these two pain models and assayed the resulting GFAP immunostaining in SGCs in DRG. In control mice, only ~1% of the neurons were surrounded with GFAP-positive SGCs, and SNL increased this to a peak value of about 80% after one day, which declined over the following days. The local application to the nerve of the Na^+^ channel blocker tetrodotoxin from a diffusion pump decreased this value to about 20% on day 1. In SNI rats, the percentage of GFAP-surrounded neurons peaked at nearly 60% on days 3–6, and a local application of the local anesthetic bupivacaine reduced this value to 20–25%. It was concluded that blocking nerve conductance reduced GFAP staining substantially, but not fully. The authors suggested that the injury induced strong firing in the nerves that was conducted to the somata of sensory neurons, which in turn transmitted signals to the SGCs, resulting in GFAP synthesis in them. There was a component of GFAP upregulation that was independent of nerve conduction, but even so, these findings imply that firing in DRG neurons contributes to SGCs’ activation. The long delay between injury and the GFAP upregulation is not easy to explain but may be due to late spontaneous firing in the DRG, which can be observed several days after the injury [17,18]. The disagreement with the results by Romero et al. [9] suggests that the injury type or species (rats vs. mice) influences the results. In this context, it may be noted that, unlike in rats, GFAP was not elevated in SGCs of mice in a model of neuropathic pain [19]. In contrast, GFAP was upregulated in both species in a model of systemic inflammatory pain [19,20]. Obviously, there is a need for better, or at least additional, markers of glial activation.

As mentioned above, nerve conductance from the periphery requires less than a second (even via the slow conducting C fibers), but GFAP synthesis is likely to require a much longer time. Moreover, how the neurons signal to SGCs is not known and, of course requires some time. This process is discussed below.

### 3.2. Signaling across the Spinal Cord

In most studies on nerve injury, the injury is unilateral; for example the sciatic nerve on one side is lesioned, or one paw is injected with complete Freund’s adjuvant (CFA) to induce local inflammation [21,22]. In the majority of the studies, only the ipsilateral side showed behavioral consequences of the lesion [16,23,24,25,26]. Furthermore, the contralateral side frequently serves as the control, as there is evidence that this side is much less affected by the experimental manipulation [16,27,28,29]. This is consistent with the assumption that injury signals travel in the axons and influence only the corresponding neurons (In fact, this is a simplification because there is evidence for the spread of signals to neighboring neurons.). However, there are numerous reports that both ipsi- and contralateral DRGs were affected by unilateral lesions in a variety of animal models [7,30,31,32,33,34,35]. Currently, there is no clear evidence for functional or clinical manifestations of this concept in humans. For example, postherpetic neuralgia pain is limited to the ipsilateral side [33]. A possible explanation for these observations is that neural signals can cross the spinal cord and activate neurons contralaterally to the injury [33]. This explanation is plausible in principle, but the possible underlying mechanisms are obscure. It should be noted that the changes on the contralateral side are similar to those on the ipsilateral one, which would require an exact mapping of the changes across the spinal cord.

Several authors reported that unilateral injury can cause bilateral pain and referred to it as “mirror image pain”, which was explained by signal transmission across the spinal cord. Spataro et al. [36] suggested that gap junctions involving spinal glia underlay this effect. This was supported by Choi et al. [37], who found that lowering the expression of the gap junction protein Cx43 in spinal astrocytes reduced mirror image pain in rats.

Dubový and his coworkers described both contralateral and long-range changes resulting from unilateral sciatic nerve injury in rats. They found that unilateral nerve injury increased the levels of the cytokines IL-6, TNF-α and IL-10 not only in the associated lumbar DRG bilaterally, but also in the cervical C7-8 DRG on both sides [31,38]. The hind paws of rats that underwent sciatic nerve injury displayed decreased withdrawal thresholds for mechanical allodynia and thermal hyperalgesia, but no significant behavioral changes were found in the contralateral hind paw and both forepaws. Thus, the biochemical observations at the contralateral lumbar DRG and C7-8 level were not accompanied by behavioral manifestations. The authors explained these results by neuronal interactions within the spinal cord, as suggested by Koltzenburg et al. [33], or by humoral spread (see below).

Together, these observations suggest non-local effects following focal injury; however, further verification and clarification of the underlying mechanisms are needed.

### 3.3. Axonal Transport

Axonal transport is an essential means for transporting proteins, neurotransmitters, and other molecules from the cell body to the axon and its terminals (anterograde or orthograde direction). Transport can take place in the opposite direction as well (retrograde), which is more relevant to the present discussion. Two main types of axonal transport are known, slow and fast. Slow axonal transport occurs at a rate of 0.1–10 mm/day [39,40], which rules out the contribution of slow axonal transport to the fast ganglionic changes. However, the rate of fast axonal transport in rodent peripheral nerves can be as high as 13–14 mm/h [39,40,41,42], and considering that in rodents the distance between terminals and ganglia is of the order of several cm, this mechanism may mediate fast changes in the ganglia because it may enable the movement of molecules from the site of injury to sensory ganglia within 1–2 h. Still, the delay of 2–10 min mentioned above requires a much faster transfer rate, which can only be achieved by nerve conduction. Axonal transport of molecules such as growth factor from the periphery is well known [43] and may play a role in the responses in cells of sensory ganglia to peripheral injury. In summary, it is conceivable that some events that take place in the ganglia in hours/days depend on both slow and fast transport.

### 3.4. Humoral Signaling

As mentioned above, in many studies unilateral lesions induce only ipsilateral changes in the sensory ganglia and spinal cord. However, in principle, nerve injury can have systemic consequences. Damage induces a variety of local events, such as the release of proinflammatory cytokines and other factors, which can enter the circulation and reach most body tissues, including several brain circumventricular regions that have a leaky blood brain barrier. In an attempt to explain changes in the cervical DRG due to sciatic nerve damage, Dubový et al. [31] measured the plasma level of IL-6 and found a 1.5–2-fold increase. However, this was not correlated temporally with the neuropathic pain observed ipsilaterally.

In a later study, Dubový et al. [44] hypothesized that injured axons released mitochondrial DNAs (mtDNA) into the bloodstream. mtDNA may act by binding to the toll-like receptor 9 (TLR9) in remote DRGs, inducing neuroinflammatory responses in them. The significance of this observation remains to be elucidated. Dubový et al. [44] listed additional hypothetical mechanisms, which need to be tested. Again, it should be noted that in cases where contralateral effects are observed, they are quite specific and are limited to the contralateral organ (e.g., right and left limbs), and not to all sensory ganglia. In contrast, systemic effects are expected to be less specific and more widespread. In summary, systemic effects of local injury are conceivable, but this topic requires much further investigation.

In conclusion, signals from the periphery can travel into the sensory ganglia by a variety of means. Some of the ideas mentioned above require further validation, and the underlying mechanisms are largely hypothetical. Currently, electrical conductance appears to be a likely possibility because there is both experimental support and a reasonable mechanism by which it operates. Moreover, electrical activity in the neurons can explain how they communicate with SGCs (see the next section). Figure 1 summarizes the current view of the author.

## 4. How Do Neurons Signal to SGC?

In this section, we will examine how changes induced in the sensory neurons can be relayed to SGCs, but first the question of neuron–neuron interactions needs to be mentioned. There is considerable evidence for “cross talk” between DRG neurons, which appears to be mediated by chemical messengers [45,46,47], but also by gap junctions [21,48]. The communicating neurons are likely to be in close proximity within the ganglia [47]. In contrast, signaling from neurons to SGCs seems to take place over longer ranges. Stephenson and Byers [49] injured a single tooth in rats and noted that GFAP was upregulated not only around injured TG neurons, but also around TG neurons that innervated orofacial regions not affected by the injury and were located in other regions of the ganglion. As TG targets are mapped onto different parts of the ganglion, this makes the analysis much easier than for DRG, where no clear mapping was found. Stephenson and Byers [49] suggested that this change in SGCs was induced by the release of substances from the neurons, among them CGRP and nitric oxide (NO), which influence the SGCs. This idea is supported by later work on TG [50] and DRG [51]. A likely candidate for the chemical messenger between neurons is ATP, acting on purinergic P2 receptors [46,47]. It has been proposed that ATP might participate in neuron-SGCs signaling by mediating calcium waves [5,52].

NO seems like an attractive candidate for a messenger between neurons and SGCs, as it is a very small and short-lived molecule that can diffuse in both hydrophilic and lipophilic environments. Indeed, NO was found to be released by neurons and to raise cyclic guanosine monophosphate (cGMP) synthesis in SGCs in cultured DRG [53]. We reported that local colonic inflammation induced SGC activation, as assayed by the upregulation of GFAP in SGCs of mouse DRG [54]. We tested the idea that the messenger between affected neurons and SGCs was NO, acting by elevating cGMP in SGCs. We found that incubating isolated ganglia with the NO donor sodium nitroprusside or with a cell-permeable cGMP analog caused GFAP upregulation, increased SGC coupling by gap junctions, and increased SGC responses to ATP. These treatments also raised neuronal excitability. Blocking NO synthesis in ganglia from the colonic inflammation model with the NO synthase blocker L-NAME abolished all the neuronal and glial changes listed above. These results, together with work from other laboratories, led to the following conclusions: Nerve activity in injured neurons causes the formation of NO, which diffuses to nearby SGCs. This induces cGMP in SGCs, which leads to their activation and the associated changes in them, which results in neuronal hyperexcitability and pain. Enhanced sensitivity to ATP together with augmented gap junctions enable the spread of calcium waves, which contribute to neuronal excitation. To test the hypothesis that neuronal firing triggered SGCs’ activation via NO, we carried out in vitro experiments where neurons in intact DRGs were stimulated with capsaicin [55]. This induced SGCs’ activation, which was prevented by incubation with L-NAME, thus confirming the hypothesis.

There is evidence that other mediators may also play a role in neuron–SGC communications. For example, ATP released by neurons may act on SGCs [51,55,56]. The release of cytokines from SGCs is another likely possibility [5,57,58]. It is well established that activated SGCs release cytokines such as IL-1β and TNF-α [5,32,57,58]. These mediators can easily diffuse to neighboring neurons and increase their excitability, thus evoking nociception [58,59,60]. One can expect future work to reveal additional mechanisms.

## 5. Clinical Implications

The preceding discussion dealt with two communication processes—from the periphery to sensory neurons, and from neurons to SGCs. Both appear to be relevant to the transmission and processing of sensory signals under normal and pathological states, and in particular pain.

Post-surgical pain is observed in 10–50% of patients undergoing common operations, such as thoracotomy, hip and knee operations, and mastectomy, and it is a major public health concern [61,62]. The pain can be severe, and its treatment is a major problem. Based on animal studies on injury potential (see above), it was proposed that blocking abnormal firing in the axons of sensory ganglia may prevent or diminish post-surgical pain, the so-called “preemptive analgesia”. The rationale behind this approach is that blocking the injury potential might prevent the subsequent development of chronic pain. This idea has a sound biological basis, but despite the potential in this approach, there is currently no consensus about its usefulness and it did not gain popularity in clinical practice [63]. This has been attributed to several inherent methodological problems; it is difficult to distinguish between events that occur during surgery and those that occur after it, as both can contribute to the post-surgical pain [14]. There is considerable variability across surgical procedures in the nature and extent of tissue damage and nerve injury. Thus, various surgical procedures differentially affect variables with respect to the duration, intensity, and quality of the noxious input. As a result, the same analgesic regimen administered for different procedures could lead to different outcomes and conclusions regarding the viability of preventive analgesia. Another reason for the insufficiency of preemptive analgesia is that signals from the injury may be relayed to the sensory ganglia and the spinal cord by mechanisms other than electrical conduction, for example by axonal transport, which is not expected to be influenced by nerve blockade. This idea has received some attention. Sotigiu et al. [64] used the chronic constriction model of the sciatic nerve in rats to test the role of axonal transport in wide dynamic range (WDR) spinal neurons, which receive nociceptive signals. A local application of vincristine, which blocks fast axonal transport, reduced the sensitization of WDR neurons. The authors concluded that the retrograde transport of (unknown) substances contributed to neuronal sensitization and pain and suggested that the blockade of this transport may be used for the prevention of post-surgical pain. Devor and Govrin–Lippmann [65] obtained similar results and suggested that fast axonal transport blockers acted by reducing the anterograde transport of Na^+^ channels and/or adrenergic receptors. However, it should be noted that such a blockade may have adverse effects because it inhibits the flow of opioid peptide from DRG to the periphery, which has an analgesic influence [66]. This topic should be studied further using modern methods.

An alternative approach to the treatment and prevention of post-surgical pain may be to reduce the pathological interactions between SGCs and neurons, for example by blocking the upregulation of pERK [12] or by inhibiting NO production by sensory neurons [54]. Further options are likely to emerge as we learn more about neuron–SGC interactions in health and disease.

## Figures and Tables

**Figure 1 cells-11-00512-f001:**
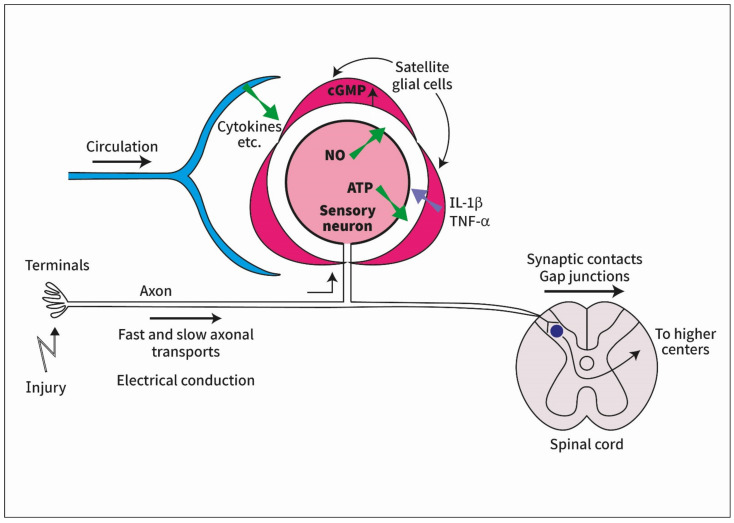
Diagram describing some of the ideas on how signals from peripheral injury reach satellite glial cells (SGCs) in sensory ganglia. Signals from the periphery travel in the axon of the sensory ganglion (by electrical conduction or axonal transport). Electrical activity in the neuronal somata induces the synthesis of nitric oxide (NO), which diffuses to SGCs and evokes the synthesis of cyclic guanosine monophosphate (cGMP) in them. This in turn induces SGC activation that leads to the release of cytokines from SGCs, which can increase neuronal excitability. Other pathways from the periphery to the ganglia include circulating substances such as cytokines and the transmission of electrical signals to contralateral ganglia across the spinal cord.

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
