# Peer review of "How Is Peripheral Injury Signaled to Satellite Glial Cells in Sensory Ganglia?"

_cells, 2022, doi:10.3390/cells11030512_

Round 1

Reviewer 1 Report

Dear author,

The review is well written, the scientific topics are disclosed in some detail. The article is of great interest to scientists of neuroscientists and other sciences. However, the author of the article should supplement the mini-review with schematic data of cellular and molecular neuron-SGCs interactions for clarity and a clearer understanding of complex processes.

Author Response

The review is well written, the scientific topics are disclosed in some detail. The article is of great interest to scientists of neuroscientists and other sciences. However, the author of the article should supplement the mini-review with schematic data of cellular and molecular neuron-SGCs interactions for clarity and a clearer understanding of complex processes.

Response

I thank the reviewer for the favorable opinion.

Additional information was inserted to Figure 1.

Reviewer 2 Report

Manuscript ID: cells-1581540

Type of manuscript: Review

Title: How is peripheral injury signaled to satellite glial cells in sensory ganglia?

Authors: Menachem Hanani

The MS is a review of the activation of satellite glial cells and signaling from injured primary sensory neurons. The author provides an overview of current knowledge on pathways from injured peripheral nerves to sensory ganglia as well as proper neuronal signals to the satellite glial cells based mainly on the author’s published results.

The text of the Review requires additions and formal adjustments before publication.

The author should pay more attention to the text, especially:

Line 196-197: Fig. 1 summarizes the current view of the author. However, the figure is missing in the submitted MS.

Some sentences need English correction: e.g., lines 56-57, 64-65, 231-232, 246 (Post-surgical pain)

The author should reconcile the statements about ATP involvement in the activation of SGCs -lines 212-214 and 238-240

Lines 33-34 It is not true that: …… astrocyte activation and slow changes in microglia. It should be reworded if it means sensory afferentation.

It was clearly demonstrated that peripheral nerve injury induces an early spinal microglial activation that precedes astrocytic activation (Raghavendra et al. 2003; Zhang and Koninck 2006; McMahon and Malcangio 2009). The delayed but sustained activation of astrocytes indicates that they are implicated in the maintenance phase of neuropathic pain (Tanga et al. 2004). Generally, microglial cells are activated at the initial stages of nerve injury, their activation is reduced with time and it is followed by late activation of astrocyte continuing during the later periods. 

Author Response

I thank the reviewer for the useful comments.

I believe that I had uploaded Figure 1. I made sure that this was done now.

Some sentences need English correction: e.g., lines 56-57, 64-65, 231-232, 246 (Post-surgical pain)

Response

All these comments were corrected, and I thank the reviewer for noticing the mistakes.

The author should reconcile the statements about ATP involvement in the activation of SGCs -lines 212-214 and 238-240

Response

This is probably a misunderstanding; ATP is a messenger between neurons and SGCs,  but the main factor inducing SGC activation is nitric oxide, not ATP.

Lines 33-34 It is not true that: …… astrocyte activation and slow changes in microglia. It should be reworded if it means sensory afferentation.

It was clearly demonstrated that peripheral nerve injury induces an early spinal microglial activation that precedes astrocytic activation (Raghavendra et al. 2003; Zhang and Koninck 2006; McMahon and Malcangio 2009). The delayed but sustained activation of astrocytes indicates that they are implicated in the maintenance phase of neuropathic pain (Tanga et al. 2004). Generally, microglial cells are activated at the initial stages of nerve injury, their activation is reduced with time and it is followed by late activation of astrocyte continuing during the later periods. 

Response

I agree that this sentence was badly worded. It has been corrected as: “ ..early microglia activation, followed by a late astrocyte activation.”

Reviewer 3 Report

The review “How is peripheral injury signaled to satellite glial cells in sensory ganglia” by Hanani is written clearly and reviews important mechanisms of periphery-DRG-signaling emphasizing the role of satellite glial cells (SGCs).

There are only some minor points, I would recommend to revise:

1) For me it could be more clear, in how far SGC-activation affects neuronal excitability (e.g., via cytokines, chemokines). Even if the focus of this review is on the communication from periphery via DRG neurons to SGCs, this point is of significance when a role of SGCs for pain hyperexcitability is discussed. However, it is only mentioned briefly (page 5, line 229-231).

2) When discussing humoral signaling pathways (page 4, line 171), the author mentions the “exception of the CNS, which is protected by the blood brain barrier”. At this point, I would emphasize that there are certain brain regions (circumventricular organs, CVOs) with a leaky blood brain barrier, which have important functions in sickness behavior, e.g., by detecting circulating cytokines [see: Roth et al. 2004: “Signaling the brain in systemic inflammation: role of sensory circumventricular organs”, DOI: 10.2741/1241). Increased circulating cytokine levels, such as plasma IL-6 (line 178), may affect the brain and pain-related behavior via CVOs.

3) At several points, I found minor mistakes in spelling/grammar and would recommend to proofread the manuscript carefully:

  1. page 2, line 56: “Therefore, is can be assumed…” = “it”
  2. page 2, line 62: “…present in SGC…” = “SGCs”
  3. page 2, line 65: “pERK of in…” = “in”
  4. page 2, line 87: “lead to activation SGCs” = “of SGCs”
  5. page 3, line 111: “markers or glial activation” = “of”
  6. page 3, line 147: “Thus the…” = “Thus, the…”
  7. page 5, line 217-218: “hydrophilic and hydrophilic” = I guess one is lipophilic?
  8. page 5, line 246: “Post-surgical is observed” = I guess it’s about “pain”?
  9. page 6, line 265: “to be influences…” = “influenced”
  10. page 6, line 275: “…noted such blockade” = “that such blockade”
  11. page 6, line 275: “it inhibit…” = “inhibits”

4) Moreover, some formatting issues seem to be occurred: At several points brackets are not equally: (] (e.g., page 1, line 10 (SGCs] or line 32 [DRG] and (TG).

Author Response

The review “How is peripheral injury signaled to satellite glial cells in sensory ganglia” by Hanani is written clearly and reviews important mechanisms of periphery-DRG-signaling emphasizing the role of satellite glial cells (SGCs).

There are only some minor points, I would recommend to revise:

1) For me it could be more clear, in how far SGC-activation affects neuronal excitability (e.g., via cytokines, chemokines). Even if the focus of this review is on the communication from periphery via DRG neurons to SGCs, this point is of significance when a role of SGCs for pain hyperexcitability is discussed. However, it is only mentioned briefly (page 5, line 229-231).

Response

I thank the reviewer for the useful comments.

This point is now expanded, and two new references (Binsthok et al., 2008; Takeda et al., 2009, 59,60 ) were added.

2) When discussing humoral signaling pathways (page 4, line 171), the author mentions the “exception of the CNS, which is protected by the blood brain barrier”. At this point, I would emphasize that there are certain brain regions (circumventricular organs, CVOs) with a leaky blood brain barrier, which have important functions in sickness behavior, e.g., by detecting circulating cytokines [see: Roth et al. 2004: “Signaling the brain in systemic inflammation: role of sensory circumventricular organs”, DOI: 10.2741/1241). Increased circulating cytokine levels, such as plasma IL-6 (line 178), may affect the brain and pain-related behavior via CVOs.

Response

This was corrected, by adding “including several brain circumventricular regions that have a leaky blood brain barrier”.

3) At several points, I found minor mistakes in spelling/grammar and would recommend to proofread the manuscript carefully:

  1. page 2, line 56: “Therefore, is can be assumed…” = “it”
  2. page 2, line 62: “…present in SGC…” = “SGCs”
  3. page 2, line 65: “pERK of in…” = “in”
  4. page 2, line 87: “lead to activation SGCs” = “of SGCs”
  5. page 3, line 111: “markers or glial activation” = “of”
  6. page 3, line 147: “Thus the…” = “Thus, the…”
  7. page 5, line 217-218: “hydrophilic and hydrophilic” = I guess one is lipophilic?
  8. page 5, line 246: “Post-surgical is observed” = I guess it’s about “pain”?
  9. page 6, line 265: “to be influences…” = “influenced”
  10. page 6, line 275: “…noted such blockade” = “that such blockade”
  11. page 6, line 275: “it inhibit…” = “inhibits”

Response

All these comments were corrected, and I thank the reviewer for noticing the mistakes.

4) Moreover, some formatting issues seem to be occurred: At several points brackets are not equally: (] (e.g., page 1, line 10 (SGCs] or line 32 [DRG] and (TG).

Response

The brackets were corrected.